# Selection on mutators is not frequency-dependent

Yevgeniy Raynes*, Daniel Weinreich

Department of Ecology and Evolutionary Biology, Center for Computational Molecular Biology, Brown University, Providence, United States

**Abstract** The evolutionary fate of mutator mutations – genetic variants that raise the genome-wide mutation rate – in asexual populations is often described as being frequency (or number) dependent. Mutators can invade a population by hitchhiking with a sweeping beneficial mutation, but motivated by earlier experiments results, it has been repeatedly suggested that mutators must be sufficiently frequent to produce such a driver mutation before non-mutators do. Here, we use stochastic, agent-based simulations to show that neither the strength nor the sign of selection on mutators depend on their initial frequency, and while the overall probability of hitchhiking increases predictably with frequency, the per-capita probability of fixation remains unchanged.

## Introduction

Mutator alleles have been found at considerable frequencies in populations of infectious and commensal bacteria (*Matic et al., 1997*; *LeClerc et al., 1996*; *Oliver et al., 2000*), viruses (*Suárez et al., 1992*), and pathogenic fungi (*Healey et al., 2016*; *Billmyre et al., 2017*; *Boyce et al., 2017*). Mutators are also believed to be widespread in many cancers (*Loeb, 2011*; *Lengauer et al., 1997*), and have been repeatedly observed to overtake microbial populations during laboratory evolution experiments (*Sniegowski et al., 1997*; *Shaver et al., 2002*; *Barrick et al., 2009*; *Notley-McRobb et al., 2002*; *Pal et al., 2007*; *Mao et al., 1997*; *Raynes and Sniegowski, 2014*; *Voordeckers et al., 2015*). Yet, unlike directly beneficial mutations that are favored by natural selection because they increase an organism's reproductive success (i.e., its fitness), mutator mutations generally do not appear to be inherently advantageous (*Raynes and Sniegowski, 2014*), except potentially in some viruses (*Furió et al., 2005*; *Furió et al., 2007*). Instead, mutators experience indirect selection, mediated by persistent statistical associations with fitness-affecting mutations elsewhere in the genome. As a result, mutators may invade an adapting population by hitchhiking (*Smith and Haigh, 1974*) with linked beneficial mutations even when they have no effect on fitness of their own (*Sniegowski et al., 2000*).

Whether or not mutators can successfully hitchhike to fixation has often been described as depending on the initial prevalence of mutator alleles in a population - most commonly referred to as frequency or number dependence (*Raynes and Sniegowski, 2014*; *Sniegowski et al., 2000*). This view holds that to replace the resident non-mutators, mutators must generate a beneficial mutation that escapes genetic drift and sweeps to fixation before their non-mutator competitors do. Accordingly, it has been proposed that mutators may be expected to invade (i.e., are favored by selection) only when already present in sufficient numbers to produce the successful beneficial mutation first, and lose their advantage (i.e., are disfavored by selection) when too rare to do so (reviewed in *Raynes and Sniegowski, 2014*; *Sniegowski et al., 2000*).

This frequency-dependent interpretation of mutator success has been primarily motivated by mutator dynamics observed in experimental studies of laboratory microbial populations. Most famously, in a series of pioneering experiments, Lin Chao and colleagues showed that mutator strains of the bacterium *E. coli* could supplant otherwise isogenic non-mutator strains by hitchhiking

*For correspondence:
yevgeniy_raynes@brown.edu

Competing interests: The authors declare that no competing interests exist.

with beneficial mutations when initialized above a critical threshold frequency but would decline when initialized below it (*Chao and Cox, 1983*: reproduced in Figure 1A; *Chao et al., 1983*). Since then, a similar pattern has been recapitulated in several other studies in *E. coli* and *S. cerevisiae* (*Thompson et al., 2006*; *Gentile et al., 2011*; *de Visser and Rozen, 2006*; *Le Chat et al., 2006*). Critically, a frequency-dependent framing of indirect selection on mutators implies a change in the sign or the strength of indirect selection with frequency. Here, we use stochastic, agent-based computer simulations to demonstrate that on the contrary, indirect selection on mutators is independent of frequency.

## Results and discussion

Our computer simulations (*Raynes, 2019*) model asexual populations that mimic microbial evolution experiments under generally-accepted parameter values (*Raynes et al., 2018*). *Figure 1B* shows mutator frequency dynamics in randomly chosen simulations initialized across four log-orders of starting frequency, $x_0$, which recapitulate experimental observations of the critical frequency threshold for hitchhiking reproduced in *Figure 1A*. As in *Figure 1A*, single, randomly-chosen realizations (i.e., simulation replicates) started below a threshold frequency end in mutator loss, while randomly-chosen realizations started above end in fixation (*Figure 1B*).

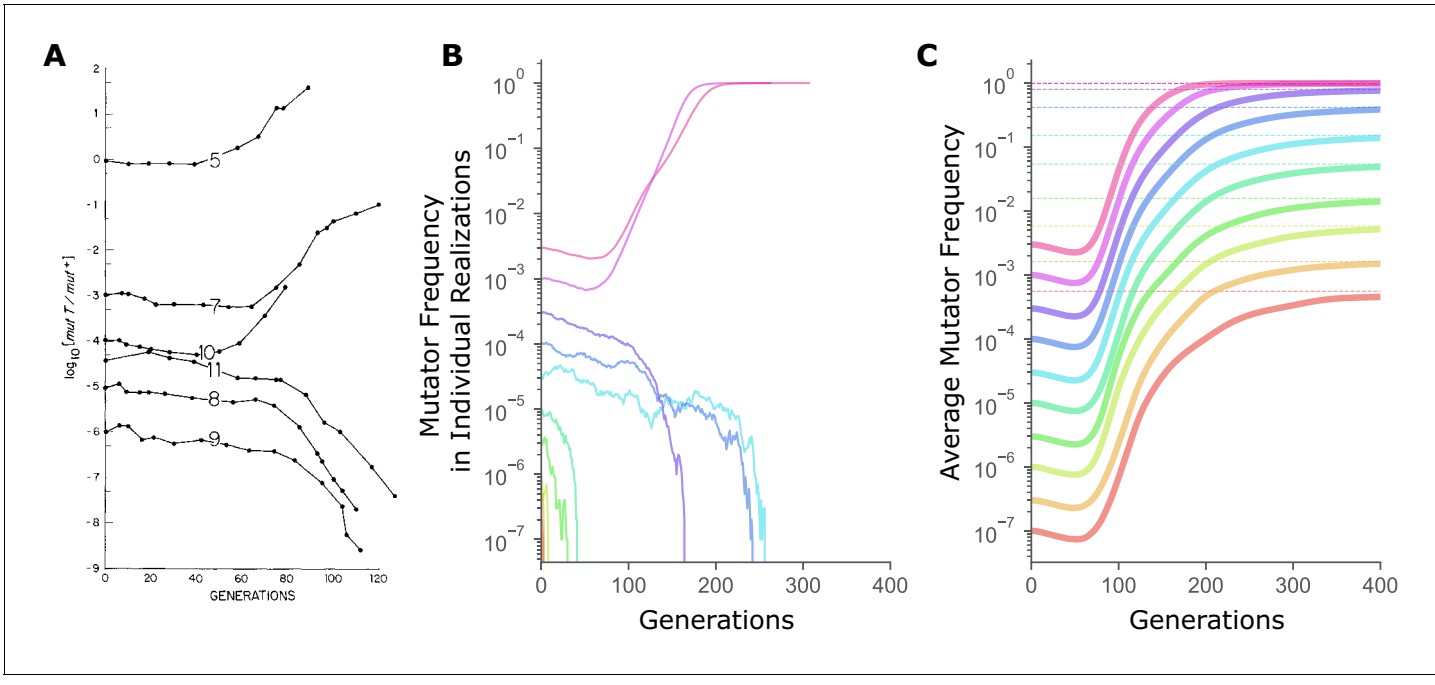

**Figure 1.** The sharp transition between fixation and loss in mutator dynamics at different starting frequencies is due to limited sampling. (A) Changes in the ratio of the mutator and the wild-type alleles of the *E. coli mutT* locus over time in continuous chemostat cultures. (Figure 1 from *Chao and Cox, 1983*). (B) In simulations, mutator trajectories in individual realizations initiated at different starting frequencies recapitulate the experimental observation of the frequency-threshold for mutator hitchhiking. Parameter values used are typical of microbial experimental populations (*Raynes et al., 2018*): $N = 10^7$, $U_{del} = 10^{-4}$, $U_{ben} = 10^{-6}$, constant $s_{ben} = 0.1$, constant $s_{del} = -0.1$. Mutators mutate 100× faster than non-mutators. (C) Average mutator trajectories across realizations do not show evidence of the frequency-threshold. On average, mutators increase in frequency at all $x_0$, showing that selection favors mutators independent of frequency. Average mutator frequency always eventually reaches the expected $P_{fix}(x_0)$ (dashed horizontal lines) calculated in *Figure 2*. Mutator frequencies averaged across $10^6$ simulation runs at $x_0 = 10^{-7}$ and $x_0 = 3 \times 10^{-7}$, and across $10^5$ simulation runs for all other starting frequencies. For simulations with exponentially distributed selection coefficients see *Figure 1—figure supplement 1*.

The online version of this article includes the following source data and figure supplement(s) for figure 1:

**Source data 1.** Numerical data represented in *Figure 1*.

**Figure supplement 1.** Simulations with exponentially distributed selection coefficients confirm that the frequency-dependent threshold in mutator dynamics is due to limited sampling.

**Figure supplement 1—source data 1.** Numerical data represented in *Figure 1—figure supplement 1*.

Critically, fixation of an allele in a finite population is a probabilistic process influenced both by selection and random genetic drift, and even beneficial mutations will frequently be lost by chance alone. As such, whether an allele is truly favored or disfavored by selection can only be ascertained by evaluating its expected behavior averaged across many replicate, independent realizations. Indeed, if we consider the expected mutator frequency averaged across many replicate simulations, the threshold-frequency effect disappears (*Figure 1C*). Instead, the average mutator frequency ultimately rises above the starting frequency at all $x_0$, suggesting that mutators are, in fact, favored by selection in these populations regardless of starting frequency. (For more on why mutators are favored in large populations such as these see *Raynes et al., 2018*). The transient decline in average frequency seen in *Figure 1C* reflects selection against the deleterious load inherent to mutators (*Kimura, 1967*), and will be explored in a forthcoming publication].

To confirm that selection on mutators is independent of starting frequency, we measured the fixation probability of a mutator allele, $P_{fix}(x_0)$, at each initial frequency, $x_0$ simulated in *Figure 1*. Given the stochasticity of the fixation process (and following *Good and Desai, 2016*; *Raynes et al., 2018*; *Wylie et al., 2009*), we compare $P_{fix}(x_0)$ to the probability of fixation of a neutral allele, given simply by $x_0$. If a mutator is favored, we expect it to fare better than neutral (i.e., $P_{fix}(x_0)>x_0$), and worse than neutral (i.e., $P_{fix}(x_0)<x_0$) if disfavored. As *Figure 2* shows, $P_{fix}(x_0)$ exceeds the fixation probability of a neutral allele for all $x_0$, as anticipated in *Figure 1C* and confirming that the sign of selection on mutators does not depend on starting frequency.

Furthermore, while $P_{fix}(x_0)$ of a mutator increases with $x_0$, it does so exactly as expected for a frequency-independent mutation. Under frequency-independent selection $P_{fix}(x_0)$ is simply the probability that at least one of the $x_0N$ alleles reaches fixation (where $N$ is the population size). By definition of frequency-independent selection, the per-capita fixation probability is a constant, written, $P_{fix}(x_0 = 1/N)$. Correspondingly, $P_{fix}(x_0)$ for any $x_0$ can be calculated as

$$P_{fix}(x_0) = 1 - \left(1 - P_{fix}(x_0 = 1/N)\right)^{x_0N} \tag{1}$$

As the orange line in *Figure 2* shows, $P_{fix}(x_0)$ calculated with *Equation 1* is indistinguishable from $P_{fix}(x_0)$ observed in simulations, confirming that the per-capita fixation probability is independent of $x_0$ and equal to $P_{fix}(x_0 = 1/N)$ at any $x_0$. Thus, while the expected fixation probability of a mutator increases with $x_0$, the per-capita fixation probability remains unchanged, confirming that individual mutators do not become more likely to hitchhike to fixation when present at higher frequencies in a population.

Why then do mutators in experimental populations appear destined to go extinct when initially rare (e.g. *Chao and Cox, 1983*)? Given that this behavior has been documented across different systems and selective environments (as well as in our stochastic simulations in *Figure 1B*), it seems unlikely to depend on any shared biological property of the experimental systems. Consider, however, that the per-capita fixation probability of a mutator is relatively low – in our simulations, operating under realistic parameter values, $P_{fix}(x_0 = 1/N) = 5.6 \times 10^{-4}$. Thus even when mutators are favored, most experimental replicates with rare mutators are expected to end in mutator extinction, and only those started at frequencies higher than roughly $1/\left[NP_{fix}(x_0 = 1/N)\right]$ are expected to end mostly with mutator fixation. Considering only a single or even a few realizations at each starting frequency (as in *Figure 1A or B*) would, most likely, result in observing only the most expected outcome for each $x_0$. Indeed, all experimental studies that have documented the frequency-based transition included only a few populations at each starting frequency. Such limited sampling across a broad range of starting frequencies in these experiments would explain the sharp transition between fixation at high frequencies and loss at lower ones even when selection is frequency-independent (see also *Tanaka et al., 2003*). We expect that observing the dynamics in *Figure 1C* would be possible with more experimental replication, which, however, may not always be experimentally feasible.

In fact, the critical frequency-dependent transition observed in *Figure 1A and 1B* is not unique to mutators. Recall that $P_{fix}(x_0)$ of any mutation not under frequency-dependent selection, nevertheless, increases with starting frequency, $x_0$ (*Equation 1*). For example, even for a directly beneficial mutation, the probability of fixation from low frequencies is relatively low (*Figure 3A* Inset), Accordingly, as *Figure 3A* illustrates, single realizations of the dynamics of a directly beneficial mutation also exhibit a threshold-like switch from fixation to loss. In contrast, expected frequency dynamics averaged across many independent realizations confirm that beneficial mutations are favored by

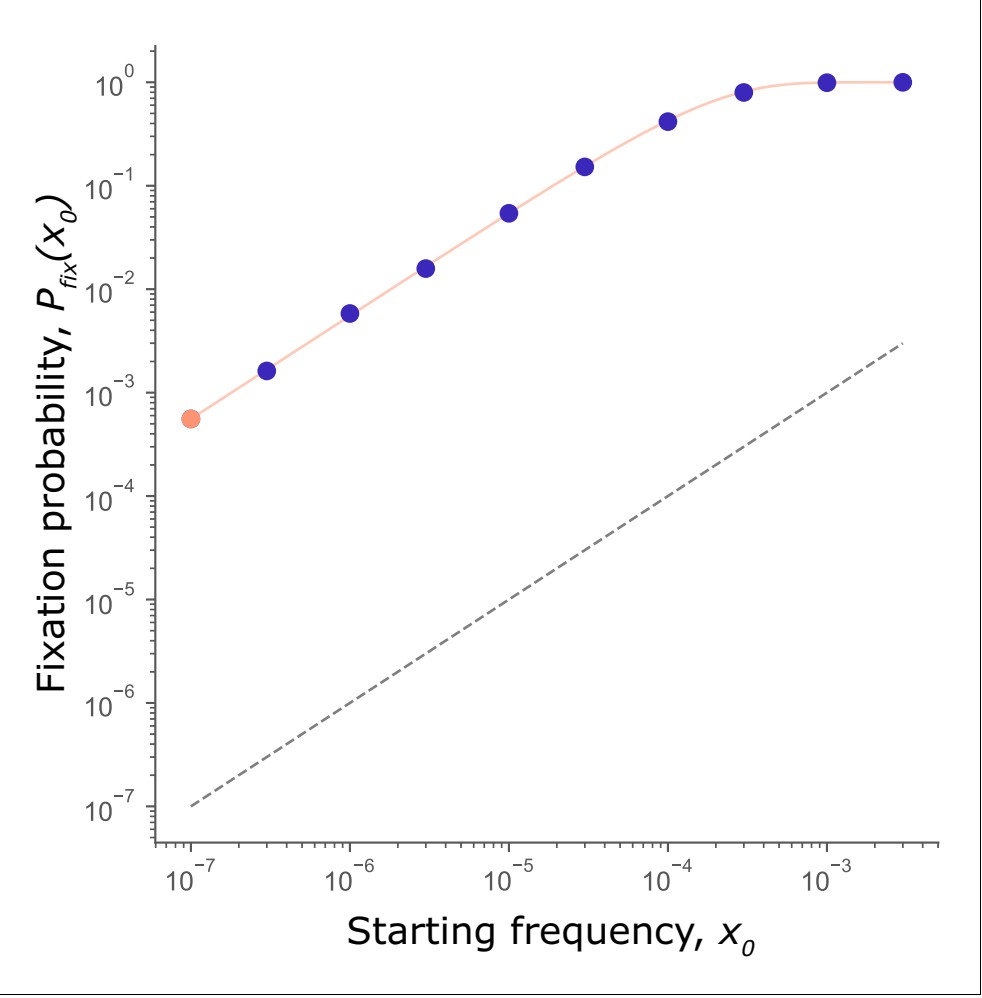

**Figure 2.** Mutator fixation probability is not frequency-dependent. Fixation probability, $P_{fix}(x_0)$, of a mutator initiated at frequency $x_0$ (circles: orange for $x_0 = 1/N$, purple for $x_0 > 1/N$). Data from simulations in **Figure 1**. $P_{fix}(x_0)$ scales with but never crosses the fixation probability of a neutral mutation ($x_0$; black dashed line). Thus, mutators are favored at all starting frequencies. The expected fixation probability $P_{fix}(x_0)$ (solid orange line), calculated from the fixation probability of a single mutator, $P_{fix}(x_0 = 1/N) = 5.6\times10^{-4}$ (orange point) using **Equation 1** is indistinguishable from the $P_{fix}(x_0)$ observed in simulations, demonstrating that the per-capita fixation probability at every frequency is independent of $x_0$ and equal to $P_{fix}(x_0 = 1/N)$.

The online version of this article includes the following source data for figure 2:

**Source data 1.** Numerical data represented in **Figure 2**.

---

selection independent of starting frequency (**Figure 3B**). Indeed, only for mutations under truly frequency-dependent selection do both the individual realizations (**Figure 3C**) and the expected dynamics averaged across many realizations (**Figure 3D**) exhibit an actual frequency-dependent transition.

In summary, our results demonstrate that neither the strength nor the sign of selection on mutators depend on initial frequency or number. Instead, we show that in populations favoring higher mutation rates, mutators consistently fare better than the neutral expectation (**Figure 1** and **Figure 2**) regardless of starting frequency. Most importantly, the per-capita probability of fixation remains unchanged with frequency. We conclude that the frequency threshold observed in earlier experiments is, therefore, an artifact of limited experimental sampling rather than a frequency-dependent change in selective effect.

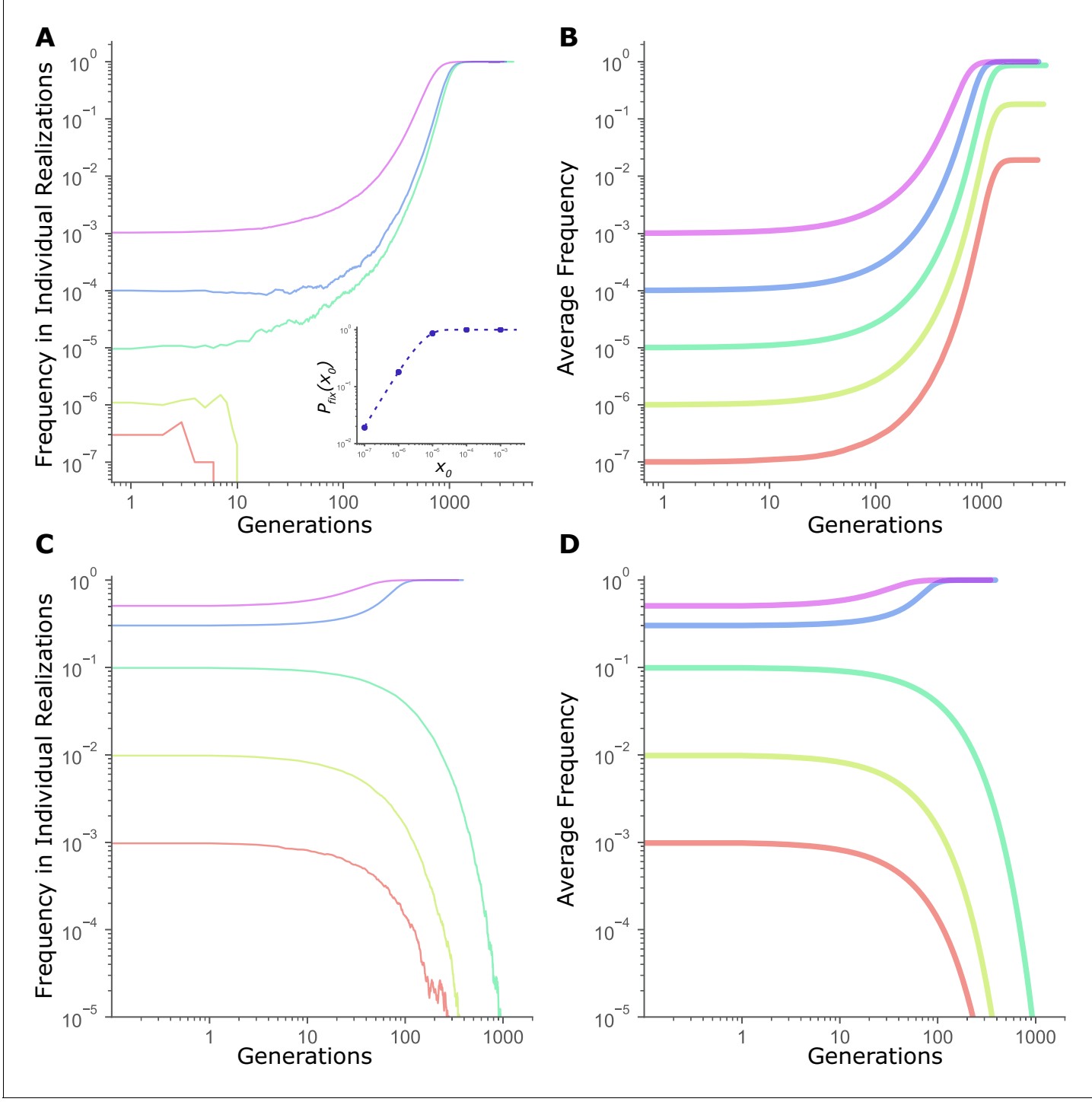

**Figure 3.** Frequency threshold in dynamics of fitness-affecting mutations. (**A**) Individual realizations of a simulation initiated with a directly beneficial mutation of size $s_{ben}$ = 0.01 at a starting frequency $x_0$. Population size, $N = 10^7$. Inset: Fixation probability of a beneficial mutation of size $s_{ben}$ =0.01 at $N$ = $10^7$. Dashed line is given by $P_{fix}^{ben}(x_0) = \frac{1-e^{-2s_{ben}Nx_0}}{1-e^{-2s_{ben}N}}$ (**Kimura, 1962**), while circles are values of $P_{fix}^{ben}(x_0)$ measured in simulations (averaged across $10^5$ runs). (**B**) Average frequency trajectories of a beneficial mutation of size $s_{ben}$ = 0.01 in (**A**) averaged across all $10^5$ runs of simulation. (**C**) Individual realizations of a simulation initiated with a mutation under frequency dependent selection, with the selection coefficient $s(x) = b + mx$, where $x$ is the frequency, $b$ = -0.02, and $m$ = 0.1, at $N = 10^7$. (**D**) Average frequency trajectories of the frequency-dependent mutation in (**C**) averaged across all $10^5$ runs of simulation. All panels are on a log-log scale for clarity.

The online version of this article includes the following source data for figure 3:

**Source data 1.** Numerical data represented in **Figure 3**.

## Materials and methods

Individual-based, stochastic simulations employed here have been previously described (*Raynes et al., 2018*). In brief, we consider haploid asexual populations of constant size, *N*, evolving in discrete, non-overlapping generations according to the Wright-Fisher model (*Ewens, 2004*). Populations are composed of genetic lineages - subpopulations of individuals with the same genotype. A genotype is modeled as an array of 99 fitness-affecting loci and 1 mutation rate modifier locus, which in a mutator state raises the genomic mutation rate *m*-fold. For computational efficiency, simulations in *Figure 1* assume constant fitness effects: beneficial mutations at the fitness loci increase fitness by a constant effect $s_{ben}$, while deleterious mutations decrease fitness by a constant effect $s_{del}$. We assume additive fitness effects and so calculate fitness of a lineage with *x* beneficial and *y* deleterious mutations as $w_{xy} = 1 + xs_{ben} - ys_{del}$. In simulations in *Figure 1—figure supplement 1*, beneficial and deleterious fitness effects are randomly drawn from an exponential distribution with the mean $s_{ben}$ = 0.1 and $s_{del}$ = -0.1 respectively. Simulations start with the mutator allele at a frequency of $x_0$ and continue until it either fixes or is lost from a population.

Every generation the size of each lineage *i* is randomly sampled from a multinomial distribution with expectation $N \cdot f_i \cdot \left( w_i / \bar{w} \right)$, where $f_i$ is the frequency of the lineage in the previous generation, $w_i$ is the lineage's fitness, and $\bar{w}$ is the average fitness of the population ($\bar{w} = \sum f_i \cdot w_i$). Upon reproduction, each lineage acquires a random number of fitness-affecting mutations *M*, drawn from a Poisson distribution with mean equal to the product of its size and its total per-individual mutation rate, ($U_{ben} + U_{del}$), where $U_{ben}$ and $U_{del}$ are the deleterious and beneficial mutation rates respectively. The number of beneficial and deleterious mutations is then drawn from a binomial distribution with *n=M* and $P = U_{ben}/(U_{ben} + U_{del})$ and new mutations are assigned to randomly chosen non-mutated fitness loci.

## Data availability

All simulated data were generated in Julia 1.0. Simulation code is available under an open source MIT license at https://github.com/yraynes/Mutator-Frequency (*Raynes, 2019*; https://github.com/elifesciences-publications/Mutator-Frequency).

## Acknowledgements

We thank Paul Sniegowski, Lin Chao, and Benjamin Galeota-Sprung for comments on the manuscript. Simulations were performed on the computing cluster of the Computer Science Department at Brown University. The work was supported by National Science Foundation Grant DEB-1556300.

## Additional information

### Funding

| Funder | Grant reference number | Author |
|---|---|---|
| National Science Foundation | DEB-1556300 | Yevgeniy Raynes Daniel Weinreich |

The funders had no role in study design, data collection and interpretation, or the decision to submit the work for publication.

### Author contributions

Yevgeniy Raynes, Conceptualization, Software, Investigation, Writing—original draft, Writing—review and editing; Daniel Weinreich, Conceptualization, Supervision, Funding acquisition, Project administration, Writing—review and editing

### Author ORCIDs

Yevgeniy Raynes (iD) https://orcid.org/0000-0003-1608-9479
Daniel Weinreich (iD) http://orcid.org/0000-0003-1424-7541

Decision letter and Author response
Decision letter https://doi.org/10.7554/eLife.51177.SA1
Author response https://doi.org/10.7554/eLife.51177.SA2

## Additional files

### Supplementary files
• Transparent reporting form

### Data availability
Simulation code is available at https://github.com/yraynes/Mutator-Frequency (copy archived at https://github.com/elifesciences-publications/Mutator-Frequency).

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
