## [Decision Letter]

**Acceptance summary:**

Mutator genotypes have elevated mutation rates, introducing more deleterious, neutral, and beneficial mutations into a population each generation than ("wild-type") non-mutator genotypes. Prior work has shown that such mutator genotypes are unlikely to become fixed within a population when they exist at low frequencies. This study challenges that view and advances our understanding of evolutionary processes by showing that the probability of mutator genotypes fixing within a population are not dependent upon their starting frequency in the population. The authors argue and then show via many simulations that the probability of fixation does not change with starting frequency under the parameter space of a typical experimental evolution study.

**Decision letter after peer review:**

Thank you for submitting your article "Selection on mutators is not frequency-dependent" for consideration by *eLife*. Your article has been reviewed by three peer reviewers, and the evaluation has been overseen by Patricia Wittkopp as the Senior and Reviewing Editor. The following individual involved in review of your submission has agreed to reveal their identity: Timothy Cooper (Reviewer #3).

The reviewers have discussed the reviews with one another and the Reviewing Editor has drafted this decision to help you prepare a revised submission.

Summary:

This study shows that the per-capita probability of fixation of mutator genotypes, including the beneficial, neutral or deleterious mutations they produce, are not dependent on starting frequency. This goes against conventional wisdom because experiments with mutator genotypes of microbes have indicated that mutators fail to invade from low starting frequencies, i.e., below the level of their relative increases in beneficial mutation supply rate. The authors argue and then show via many simulations that the probability of fixation does not change with starting frequency under the parameter space of a typical experimental evolution study. This is a notable finding.

Essential revisions:

1) The primary concern raised (and supported by all three reviewers) is that the paper presents a straw man argument. As the authors note numerous times in the manuscript, the argument that selection for mutators is dependent on frequency is "common intuition". But the reviewers were not convinced that this intuition is widespread. They thought most people would agree that the probability of fixation is related to mutator frequency, but not that the selective coefficient/advantage for a given mutator changes with frequency in a population.

Basic evolutionary theory would suggest that if the selective advantage of a mutator is due to its linkage with a beneficial mutation, then this selective advantage will be related to the probability of causing a beneficial mutation (i.e., the ratio of beneficial and deleterious mutation rates). This should be independent of how many mutators there are in a population. Of course the probability that a mutator genotype will fix increases with the frequency of mutators. This is simply true for neutral and beneficial mutations in finite populations. In the case of a mutator genotype, it is also true because there is an increase of sampling of mutations and the probability that one of the mutators in the population hits on a beneficial mutation to hitchhike with. But the selective advantage of any individual mutator should not change. I guess what I am saying is that results of the paper seem to demonstrate what a careful consideration of mutators would suggest and that this research is not debunking some wide-held view. That being said, it is a clear and thoughtful demonstration of the phenomenon, which apparently has not been published before, and may contribute to the interpretation of experimental results.

2) A similar straw-man-like concern: Regarding relevance – the argument that the authors who have found evidence of frequency-dependence (Chao and Cox, 1983, Thompson et al., 2006, Gentile et al., 2011, de Visser and Rozen, 2006, Le Chat et al., 2006) simply did not sample *enough* to find evidence of frequency-independence is not satisfying because the pattern has replicated across systems and laboratories. This suggests there's more to the biological story here that your simulations fail to capture. What might this be? Please add a paragraph proposing reasons for this disconnect. For instance, could this be a property of the DFE, where one particular type of highly beneficial mutation trumps all, and the lucky genotype that acquires this mutation becomes destined to fix? Could this be a property of the degree of clonal interference or the effective population size, both of which alter drift?

3) About the methods: It seems as though the way the simulations are set up, all deleterious and all beneficial mutations have the same fitness cost/benefit. Is there a reason why the authors didn't draw from a distribution of fitness effects? I am not sure how much changing this would influence the results, but it does seem more biologically realistic. Perhaps commenting on this approach of simulations and parameters would be helpful.

---

## [Author Response]

Essential revisions:1) The primary concern raised (and supported by all three reviewers) is that the paper presents a straw man argument. As the authors note numerous times in the manuscript, the argument that selection for mutators is dependent on frequency is "common intuition". But the reviewers were not convinced that this intuition is widespread. They thought most people would agree that the probability of fixation is related to mutator frequency, but not that the selective coefficient/advantage for a given mutator changes with frequency in a population.Basic evolutionary theory would suggest that if the selective advantage of a mutator is due to its linkage with a beneficial mutation, then this selective advantage will be related to the probability of causing a beneficial mutation (i.e., the ratio of beneficial and deleterious mutation rates). This should be independent of how many mutators there are in a population. Of course the probability that a mutator genotype will fix increases with the frequency of mutators. This is simply true for neutral and beneficial mutations in finite populations. In the case of a mutator genotype, it is also true because there is an increase of sampling of mutations and the probability that one of the mutators in the population hits on a beneficial mutation to hitchhike with. But the selective advantage of any individual mutator should not change. I guess what I am saying is that results of the paper seem to demonstrate what a careful consideration of mutators would suggest and that this research is not debunking some wide-held view. That being said, it is a clear and thoughtful demonstration of the phenomenon, which apparently has not been published before, and may contribute to the interpretation of experimental results.

We agree that our conclusion that selection on mutators is not frequency-dependent could be arrived at by a “careful consideration” of the evolutionary forces acting on them. However, as we now further emphasize in the paper, mutator dynamics have been repeatedly described as being frequency- or number-dependent in the literature (Introduction, last paragraph). By definition this means that the fitness of an individual mutator allele would depend on its frequency and not only that the probability of fixation increases with frequency (which is true for a frequency-independent allele as well). The goal of our paper is to clarify this feature of selection on mutators by providing a clear demonstration of frequency independence.

We also appreciate the concern that while we present this misconception as a “common intuition,” it may not be as widely held as we may think. In the revision we have attempted to be more careful about not mischaracterizing the extent to which this intuition has dominated the field. We instead reframe our description of this issue in the mutator literature, especially in the experimental literature (Abstract and Introduction, last two paragraphs), where it has been advanced in a number of very influential papers. Thank you for helping us to more accurately focus this point.

2) A similar straw-man-like concern: Regarding relevance – the argument that the authors who have found evidence of frequency-dependence (Chao and Cox, 1983, Thompson et al., 2006, Gentile et al., 2011, de Visser and Rozen, 2006, Le Chat et al., 2006) simply did not sample enough to find evidence of frequency-independence is not satisfying because the pattern has replicated across systems and laboratories. This suggests there's more to the biological story here that your simulations fail to capture. What might this be? Please add a paragraph proposing reasons for this disconnect. For instance, could this be a property of the DFE, where one particular type of highly beneficial mutation trumps all, and the lucky genotype that acquires this mutation becomes destined to fix? Could this be a property of the degree of clonal interference or the effective population size, both of which alter drift?

We thank you and the reviewers for raising this point. We had not previously recognized the opportunity to discuss the difference in experimental organisms and environments in our interpretation of mutator dynamics. You and the reviewers wonder if the explanation for the pattern could be biological. But on the contrary, for us the fact that this pattern has been observed in different systems and environments *strengthens* our claim of a single, theoretical explanation. Observations from diverse systems appear to preclude the possibility that any system-specific selective property of a mutator or the available DFE could be responsible for the observation, since different alleles were used in different environments. This point is developed in the fifth paragraph of the Results and Discussion. Regarding the suggestion of a dependence in DFE we would also refer to our response to point 3, below.

Likewise, on first principles we could not see how clonal interference or population size effects could explain the apparent frequency-dependent threshold in mutator dynamics observed in individual competitions, as the threshold effect disappears when many replicates of the same competition are considered. To test this question, we conducted additional simulations parameterized exactly as those in Figures 1B and 1C except the mutation rate of the non-mutators was set to 0. This eliminates any interference from the non-mutatators. As Author response image 1 shows (averaged across 5∙10^4^ simulation runs), the pattern in Figure 1B and 1C is recapitulated in the absence of interference – random individual simulation replicates (left panel) still appear to be frequency-dependent, while average dynamics confirm that mutators are favored (i.e., increase in frequency) at all starting frequencies.

We find that the most satisfying (and parsimonious) explanation for previous experimental observations is, therefore, the limited sampling, common to all experiments. Our simulations show that such limited sampling would produce the frequency-dependent switch even when selection is truly not frequency-dependent. In the revision, we now discuss the likelihood of the biological explanation for the dynamics in different systems and the fact that the limited sampling is the most parsimonious explanation for this result in the fifth and sixth paragraphs of the Results and Discussion).

3) About the methods: It seems as though the way the simulations are set up, all deleterious and all beneficial mutations have the same fitness cost/benefit. Is there a reason why the authors didn't draw from a distribution of fitness effects? I am not sure how much changing this would influence the results, but it does seem more biologically realistic. Perhaps commenting on this approach of simulations and parameters would be helpful.

We thank you and the reviewers for this suggestion. Indeed, following our earlier work (Raynes et al., 2018 and 2019) we have assumed constant fitness effects for computational efficiency (as is now described in the first paragraph of the Materials and methods) given the large size of simulated populations. To confirm that drawing fitness effects from a distribution would not impact the conclusions of our study we have conducted additional simulations and include these in the revised manuscript. In these simulations, we allow for both beneficial and deleterious to be drawn from exponential distributions (with means equal to *s_ben_* and *s_del_* values used Figure 1). Given the added computational requirements, we have reduced the size of simulated populations (now 10^6^ vs. 10^7^ in Figure 1) and the number of starting frequencies. These new simulations (now reported in the Figure 1—figure supplement 1) recapitulate our main observation – individual, randomly-chosen realization show a frequency-dependent breakpoint while average frequency dynamics are consistent with positive selection at all starting frequencies.